# Health Promotion with Long-Term Physical Activity Program for Adults with Autism Spectrum Disorder

**DOI:** 10.3390/healthcare12020247

**Published:** 2024-01-18

**Authors:** Ayelet Dunsky, Sharon Barak

**Affiliations:** 1Department of Human Movement and Sport Sciences, The Levinsky-Wingate Academic College, Netanya 4290200, Israel; ayelet@l-w.ac.il; 2Nursing Department, School of Health Sciences, Ariel University, Ariel 40700, Israel; 3Department of Pediatric Rehabilitation, The Edmond and Lily Safra Children’s Hospital, The Chaim Sheba Medical Center, Ramat-Gan 52621, Israel

**Keywords:** adults with autism spectrum disorder, physical activity, adherence to physical activity, perseverance, quality of life

## Abstract

Individuals with autism spectrum disorder (ASD) are at higher risk for developing common chronic diseases. Engagement in physical activity (PA) can prevent health issues; however, people with ASD are known to engage in lower levels of PA in comparison to their peers. This study evaluated the effect of a long-term, 12-month PA intervention on the fitness and quality of life of adults with ASD. A quantitative approach was implemented to assess participants’ fitness, functional ability, quality of life, and participation in a range of PA classes at three different time points. Qualitative data were collected via in-depth, semi-structured interviews with three participants with ASD and three staff members. A total of 34 adults with ASD (mean age 39.76 + 7.27) participated in the quantitative part of the study. Approximately 53% of the participants exhibited perseverance and conducted adequate PA each month. Significant improvements were found in one fitness component and two quality-of-life components. Factors revealed for the program’s success were the individuals’ free choice of the PA classes and supporting people and a budget that tailored the project. Policymakers who plan health promotion programs for adults with ASD should consider long-term PA programs, with freedom of choice among PA modalities and schedules.

## 1. Introduction

Individuals with autism spectrum disorder (ASD) are at higher risk for developing common chronic diseases (e.g., hypertension and diabetes), rare conditions (e.g., Parkinson’s disease, vitamin deficiency, and genetic disorders) [1,2,3] and mental health problems [4,5] compared to the general population. Engaging in regular physical activity (PA) is recommended for people who are at greater risk for suffering from decreased health conditions [6].

Despite these recommendations, however, people with ASD tend not to meet PA guidelines and are known to engage in lower levels of PA in comparison to their peers [7,8]. It is important to note that for individuals with ASD, PA has unique benefits beyond the physical and psychological benefits for the general population, such as reduced maladaptive and stereotypic ASD behaviors [9,10] and reduced anxiety and stress [11]. As such, the need for developing attractive ways of engaging adult individuals with ASD in PA is of great importance.

In a recent meta-analysis [12] that analyzed 14 case studies (13 of which had been conducted on children), the findings indicate that PA interventions were associated with positive outcomes for individuals with ASD. Additionally, the meta-analysis concluded that insufficient data currently exist for proposing optimal PA frequency, which was found to be conducted between 1 and 5 times a week for durations between 3.6 and 12 weeks in total. In an earlier meta-analysis [13] that analyzed 8 studies (that had been conducted on children with ASD), the duration of PA intervention was found to last 4–14 weeks, with one intervention even covering 9 months.

In a recent pilot study, Savage et al. [14] studied the feasibility of a 12-week supported self-management intervention to increase physical activity for adults with ASD and intellectual disability. They found that using a smartphone’s technology significantly improved PA engagement in addition to self-management strategies and coach support. Colombo-Dougovito et al. [15] found in a qualitative study that social relationships were recalled as being important to the PA experiences among adults with ASD.

According to the Transtheoretical Model [16], in order to create a behavioral change, individuals must progress through a series of five stages that occur over time: (1) precontemplation, with no intention to change behavior; (2) contemplation, becoming aware of existing problems but not taking any steps to correct them; (3) preparation, intending to take action within the next month; (4) action, attempting to modify the behavior; (5) and maintenance, working to prevent behavioral relapses. The action stage must last for a six-month period of engaging in healthy behavior before embarking on the final maintenance stage to ensure preservation of the new behavior over time. As such, PA intervention programs that last less than six months will not be able to achieve the desired goal, i.e., the long-term preservation of the behavioral change.

Considering the literature review presented above, although PA interventions are essential for adults with ASD, most studies report data about children and are short-term interventions [9,10,13]. The purpose of this study, therefore, was to evaluate the effect of a long-term PA intervention on the general fitness and quality of life among adults with ASD, while examining the participants’ adherence to the program along the 12-month intervention. Based upon the literature, our hypothesis was that taking part in a long-term PA intervention of 12 months will improve general fitness as well as quality of life among adults with ASD.

## 2. Materials and Methods

### 2.1. Participants

The PA intervention in this study was conducted in a community village where the participants live. The community village is specifically designed for people with disabilities, and accordingly, the lifestyle and activities in the place are coordinated for the residents. Three inclusion criteria were applied: (1) gender, male or female; (2) age, 20–60 years; and (3) ASD diagnosis with all levels of functioning (high functioning = no continuous support yet with social and communicational difficulties that result in prominent impairments; moderate functioning = ongoing support needed due to prominent difficulties in verbal and non-verbal social communications; and low functioning = severe difficulties in verbal and non-verbal social skills that greatly hinder normal functioning) [17]. Moreover, two exclusion criteria were applied: (1) pre-existing health conditions, relating to any underlying or current medical condition that may constitute a contraindication for participating in a PA program without medical supervision (e.g., heart disease); and (2) cognitive inabilities, relating to profound developmental intellectual disabilities (e.g., needs are conveyed via symbolic or other non-verbal communication) [18]. A total of 34 adults with ASD participated in the quantitative section of this study during July 2017–July 2018. Participants were recruited for the study by the physical activity guides in the community village as part of explanatory conversations held with them regarding the importance of engaging in physical activity for their health. During these conversations, the study was presented, and those who were interested were invited to participate in the study.

The sample size of the study was calculated using the effect sizes obtained in a previous study conducted by the authors, “Game for Life”, on a similar population using similar outcome measures. The average effect size of this study was 0.44. With such effect size, using *t*-tests (means: the difference between two dependent means) with a prior power analysis given alpha = 0.05, power = 0.80, and effect size = 0.44, a sample size of N = 34 is required. We did not add attrition to the sample size as we did not expect big attrition since all participants reside in a community village with dedicated staff members.

### 2.2. Measurements

Most of the quantitative data were gathered at 3 different time-points: (1) T0, at the onset of the program; (2) T1, after 6 months; and (3) T2, after 12 months, while demographic data and health status [except for body mass index (BMI)] were only evaluated at the beginning of the study.

#### 2.2.1. Demographic, Functional, and Health Status

Each participant completed a questionnaire that included demographic questions (age, gender, family status, number of children, education, and religiousness; specific data on socioeconomic status were not recorded) and health status questions (strokes, epilepsy, respiratory diseases, cardiovascular diseases, hematological diseases, and diabetes). ASD functional levels (high, moderate, or low) were defined for each participant according to reports provided by their social workers and based on the criteria presented for ASD severity indicated by the Israeli Society for Children and Adults with Autism [17]. Additionally, participants’ height and weight were measured, enabling calculations of their BMI [mass (kg) × height^2^ (m)].

#### 2.2.2. Physical Fitness

To examine physical endurance, power, and flexibility, three activities were measured: (1) distance covered during a two-minute walk—participants were instructed to walk as far as they could for two minutes, and their walking distance was recorded (typical walking distance mean in adults: 65–300 m) [19]; (2) standing long-jump—participants were instructed to place their feet over the edge of the sandpit, crouch down, lean forward, swing their arms backward and forward, and then jump as far forward as possible, landing with both feet in the sandpit. The start of the jump was from a static position. The jumping distance was measured from the near edge of the sandpit to the body’s first point of contact with the ground following the jump (average jump distance in adults: 170–230 cm) [20]; and (3) sit-and-reach test—participants were instructed to sit on the floor with their legs stretched out straight ahead. The soles of their feet were placed flat against the testing box, with feet slightly apart at about hip-width. The participants were then instructed to keep their knees extended and reach forward with their hands as far as possible along the box, keeping both hands on the box. A standard ruler was placed on the sit-and-reach box to standardize measurements, starting at 23 cm from the heel. Reaches short of the toes were recorded as negative scores, while reaches beyond the toes were recorded as positive scores—all were recorded in cm and rounded up or down to the nearest 0.5 cm (average distance in adults: 27–37.5 cm) [21].

#### 2.2.3. Functional Ability

The Timed Up and Go (TUG) measure was used to assess mobility, balance, walking mobility, and fall risk. Participants were asked to stand up from a chair, walk three meters, turn around, walk back to the chair, and sit down—all at a comfortable pace. Their time for completing the test was recorded. An older adult who takes ≥12 s to complete the TUG is at risk for falling [22].

#### 2.2.4. Quality of Life

The Schalock Quality of Life Questionnaire for people with intellectual and developmental disabilities was employed in this study. This questionnaire addresses 4 scales (relating to satisfaction, productivity, empowerment, and social integration), each comprising 10 items. The scores were calculated for each domain (score range 1–10) and for the complete questionnaire (score range 40–120) [23,24].

#### 2.2.5. Program Attendance

Participant’s attendance in the program (number of monthly sessions attended—at least 60 min per session) was recorded by the program coordinator and additional staff members.

### 2.3. Qualitative Assessment

In addition to performing quantitative assessments, we also conducted qualitative assessments to expand our understanding of the statistical findings and to enable greater generalization of our findings [25]. The qualitative component in this study included in-depth, semi-structured interviews that were conducted at the end of our 12-month intervention. Three staff members at the community village and three residents who participated in the study and volunteered to be interviewed took part in this session. The interviews were then examined according to the Strengths, Weaknesses, Opportunities, and Threats (SWOT) analysis in relation to the success of the PA intervention program. Based on an ecological perspective of an organization within a given social and economic environment, SWOT suggests a simple and easy technique to examine the organization’s internal strengths and weaknesses, and the opportunities and threats to its environment [25]. As such, the structured questions focused on those components. For example, the main questions asked of the staff members were: What do you consider to be the strength of your program? What are the barriers?

The analysis procedure included the following steps: First, the audio recorded interviews were transcribed. Then, each text was coded and analyzed by both authors (AD and SB) in order to improve the dependability of the analysis. The interviews were conducted as semi-structured interviews and analyzed according to a grounded theory approach. The analysis was coded with the coding of four figures observing the findings according to the SWOT model. In each of the themes, the categories corresponding to the research question were approached [21,26].

### 2.4. Procedure

This study was performed in line with the principles of the Declaration of Helsinki. Approval was granted by the Ethics Committee of The Levinsky-Wingate Academic College. The participants and their legal guardians signed a written informed consent form prior to participating in the study. Following the first assessment, each participant joined one or more of the following PA classes at their choosing: football or basketball (once a week for an hour), dancing (once a week for an hour), swimming (1–2 h a week), running (3–5 times a week for 1.5–2 h each time), or going to the gym (1–5 times a week for an hour each session). Most activities were presented at different time slots and more than once a week, providing the participants with a large range of options to choose from at their convenience. The activities took place in facilities (sports halls, swimming pool, and gym) located throughout the communal village. All were carried out under the guidance of a staff of professional sports instructors in the village.

### 2.5. Data Analysis

Normality distribution of the main continuous variables was performed using Q-Q plots. As all main variables were normally distributed, parametric statistics were used. The participants’ demographic, functional, and health status characteristics were described through descriptive statistics (mean standard deviation, range, and percentage); differences in the prevalence of categorical variables were assessed using chi-squared tests. Changes in their fitness, functional ability, and quality of life—assessed at three different time-points—were evaluated using repeated measures analysis of variance with Tukey’s range post-hoc tests. In addition, Cohen’s d-effect size (mean ∆/standard deviation average from two means) was calculated in order to examine the extent of changes from the T0 to T2. A correction for the dependence among means was conducted using the correlations between the two means following Morris and DeShon’s equation [27]. Generally, values <0.20 were considered as trivial effect sizes, between 0.20 and 0.50 as small effect sizes, between 0.51 and 0.80 as moderate effect sizes, and >0.80 as large effect sizes. Pearson correlations between T0 and T2 in each variable were also calculated.

Participant’s attendance was calculated for each of the 12 intervention months and then utilized to examined how many months the participant had attended at least 10 PA sessions that month (i.e., the number of monthly sessions attended) using the Box-and-Whisker plot. Next, descriptive statistics of the number of months with sufficient attendance (i.e., 600 min per month—as per recommendations issued by the American College of Sports Medicine [6]) were calculated and are displayed using the Box-and-Whisker plot. Moreover, the percentage of participant attendance in each of the 12 intervention months was calculated and the results for two groups, sufficient attendance vs. insufficient attendance, were compared using chi-squared tests.

Gender (males, females), religiousness (secular, traditional), functional level, and differences in the mean number of sufficient monthly attendance were compared using independent *t*-tests (for dichotomized variables) or one-way analysis of variance (for non-dichotomized variables). For the continuous variables (age, number of children, BMI, number of chronic diseases, fitness measures, functional measures, and quality of life), associations with the number of months with sufficient attendance were evaluated using Pearson correlations.

Finally, post-hoc power analysis was conducted. The purpose of this study was to evaluate the effect of a long-term PA intervention on general fitness and quality of life among adults with ASD. Therefore, in order to calculate the study’s power, effect sizes and repeated measures correlations in all the dependent variables (T0 vs. T2) were calculated. The mean effect size observed and the mean repeated measures correlations were 0.44 and 0.49, respectively. Based on these parameters, using F tests for family and ANOVA repeated measures with within-factors design, with a sample size of N = 34, one group, and three measurements, the achieved power is 0.90.

For all statistical analyses, SPSS Statistics for Windows, version 23 (SPSS Inc., Chicago, IL, USA) was used. The level of significance was set at *p* < 0.05 (2-tailed). Only power analysis was conducted using G*Power software, version 3.1.9.2.

## 3. Results

### 3.1. Quantitative Findings

Table 1 presents the participants’ demographics, functional level, and health status. Among the 34 adults with ASD who participated in the study, the mean age was 39.76 + 7.27 years, and 70.6% of the participants were male. The participants’ ASD functional level ranged from low (n = 8, 23.50%) to high (n = 11; 32.4%). While most participants were healthy, with no comorbidities, their mean BMI was 28.46 + 5.32, depicting class I obesity. For additional information, refer to Table 1.

As seen in Table 2, of the three fitness measures (two-minute walk, standing long-jump, and the sit-and-reach test), significant improvements were only seen in the two-minute walk test distance (F-ratio = 5.35, *p* = 0.001), with a large effect size from T0 to T2. No significant changes were observed in the functional measure (i.e., TUG, effect size = −007). In quality of life, from T0 to T1, participants significantly improved in “satisfaction” (F = 5.30; *p* = 0.01), with a large effect size from T0 to T1 (effect size = 0.94). In “productivity”, although no statistically significant changes were observed, the effect size from T0 to T2 was moderate (effect size = 0.67). Significant changes were also observed in the total score; however, the effect size was small (effect size = 0.49). No significant changes and trivial effect sizes were observed in “power” and “social” scores (effect size = 0.14 and 0.04, respectively).

As seen in Figure 1, the monthly attendance per participant varied greatly, ranging from 0 to 55 sessions. The mean monthly attendance for all participants was 10.89 ± 7.09. In the following stage of analysis step, the number of months with sufficient attendance was calculated, resulting in a mean number of months with sufficient attendance of 5.02 ± 4.26. This means, on average, each participant sufficiently participated in the activities in 5.02 months. However, the range of the sufficient activity months was 0 (6 participants) to 12 months (3 participants). Figure 2 depicts the summary statistics of the attendance variable.

For each month, a central box is presented, depicting the values for the 25–75 percentiles. The central horizontal line represents the median. On the vertical axis, outside values are displayed as separate points.

The middle line in each box represents the median for that month.

Figure 3 presents the percentage of participants who attended sufficient and insufficient PA out of all 34 participants. The findings show that 52.9% of the participants performed adequate PA over the entire 12-month period. Overall, in only 1 out of the 12 months—month 10—was a significant difference seen between those who completed sufficient PA (23.5%) and those who did not (76.5%) (chi-squared = 9.52; *p* = 0.002).

In the continuous variables, statistically significant associations were observed between number of months with sufficient attendance and the distance (1) covered in the two-minute walk and (2) the standing long-jump distance (r = 0.33 and 0.44, respectively; *p* < 0.05). No significant associations were observed between age, number of chronic diseases, BMI, and any quality-of-life measures. In the categorical variables, the mean number of sufficient monthly attendance among females was lower than among males (3.0 + 3.33 and 5.87 + 4.37 months, respectively). Similarly, the mean sufficient monthly attendance of participants with moderate functioning and with high functioning (4.81 + 4.30 and 6.00 + 4.47, respectively) was significantly higher than among participants with low ASD functioning (3.50 + 3.96). No significant between-group differences were observed between secular and traditional participants (*p* > 0.05; Table 3).

### 3.2. Qualitative Findings

Analysis of the interviews led to the emergence of the following themes:

#### 3.2.1. Strengths

Three themes were found for the strengths category: (1) ”it’s all about free will”; (2) belonging to the “champions league”; and (3) assimilation of activities into a routine. The first theme, ”it’s all about free will”, relates to the variety of PA courses and schedules that were offered to the participants. As explained by the program coordinator, “The fact that participants could choose their favorite activities encouraged them to attend the project’s classes”; and mentioned by one of the participants, “The system allows me to choose a different activity each day, then I can choose what I like, and also go with friends to activities we like to do together”. Next, the theme of belonging to the “champions league” was found to be a strength of the program, as some of the classes required compliance with relevant criteria. For example, participants could only join the running club if they were able to run at a certain speed and for a certain distance. This created a prestigious status for this club, with more people trying to be accepted into it. As told by one of the running club members, “I really wanted to join this group. Once I was accepted, I felt like a champion. I feel proud that I was able to be accepted onto this prestigious club”. Finally, the assimilation of activities into a routine was enabled by offering a range of time slots for the various classes. As explained by the program coordinator, “We specifically set the gym activity hours at times when members are not busy with other activities. We went to the dining room during lunch and invited the participants to come to the gym after resting following their meal. That’s how we managed to create a new daily routine which incorporates physical activity”.

#### 3.2.2. Weaknesses

Two themes were found for the weaknesses category: (1) ”it’s all about the people” and (2) ”money makes the world go round”. The first theme emerged as most PA classes offered by the program required additional manpower. As explained by the running club instructor, “In our group, there are three different levels of speed. For each level, we need additional escorts to accompany each level of runners… If escorts are unavailable at the time, this affects the quality of the training, because the faster runners have to run slower [to accommodate the slower runners]”. Next, the ”money makes the world go round” theme emerged as the project was dependent on raising adequate funds for necessary equipment and for coaches’ salaries. This was mentioned several times by the project coordinator, such as, “We presented the project to [village] management. Thankfully, one board member has the [financial] means and after the meeting, he made a donation that allowed us to purchase four new treadmills”. He also added, “The funding from the National Insurance Institute [that we received for this project] is almost finished, and I realize that I need to raise additional funds in order to fund the coaches’ salaries. Without funding, the project would not be able to exist”.

#### 3.2.3. Opportunities

One theme was found for the opportunity category, i.e., inspirational agents. These are young people who volunteer at the village and who serve as a role model for some of the participants. The volunteers could be integrated into some of the project activities to increase attendance of some of the participants. This important insight was presented by the project coordinator. For example, “When I realized that the participants on the project really like the volunteers, I asked the volunteers to join a number of classes. Indeed, the number of participants increased significantly [in these classes], and it was evident that everyone benefited from this integration”. One of the participants mentioned, “I really like [name of a volunteer], wherever he goes I’ll come after him. If he tells me he’ll be with us in a basketball class, I’ll immediately join”.

#### 3.2.4. Threats

Finally, the threats category includes two themes: (1) competition between appealing classes and (2) dependence on other people. The first theme relates to the lifestyle at the community village, where a range of activities are offered throughout the week. Some are conducted by volunteers, and there is often an overlapping between attractive courses. As mentioned by the project coordinator, “In addition to this PA project, there are volunteers who run their own classes in the afternoon, so there is serious competition. As I said, these people are significant for the group members”. Next, the dependence on other people was a serious threat for such projects, as many participants from this community village depend on the decision-making of their social workers, rather than making their own decisions. As explained by the gym instructor, “In some cases, we have to deal with other decisions made by the social workers. Sometimes I have to convince the social worker that participating in physical activity is beneficial for the members.”

## 4. Discussion

The main finding of the current study shows that the current long-term PA intervention improved the aerobic fitness and several components of quality of life of adults with ASD. In this regard, it is important to note that participants were able to choose from different types of activities. Each activity has unique social and physical fitness demands and effects. For example, football requires social interaction, which in turn might have an enhanced influence on aspects related to psychological aspects such as quality of life. In addition, football is a very demanding sport in terms of physical fitness requirements (e.g., aerobic fitness and lower extremity strength). In contrast, swimming, although also demanding in terms of physical fitness requirements, is an individual sport and therefore might have a different effect on psychological aspects such as quality of life. Despite the anticipated different effects, the overall aim of the study was to develop a feasible program with high long-term attendance, resulting in increased PA level. The current intervention that was based on participants’ free choice to select between a range of PA classes, offered at convenient time slots during the day and promoted by significant people for the studied population, led to increased PA adherence, in line with the recommended amounts suggested by global health organizations [6]. This finding greatly contributes to the literature, as it addresses adherence of adults with ASD to health-related requirements over an entire year.

Motivation or lack thereof to engage in PA is a major psychological factor that enhances or hinders adherence, with a lack of motivation being considered a significant barrier among young individuals with ASD to engage in PA [12]. Moreover, motivation to perform PA among young adults with ASD is positively associated with three basic psychological needs: autonomy, competence, and relatedness. Thus, it has been suggested that practitioners should provide an environment where young adults with ASD can enjoy close relationships with staff members and peers within programs that are based on improving autonomy and competence [28]. In the current study, the participants could choose their PA classes based on their own desire and schedule. As such, autonomy and free choice served as a basic guideline throughout the program. Similar results were found among children, indicating that the relative efficacy of a program can be influenced by the number of alternative stimuli available [29]. Moreover, our qualitative analysis indicates that participants who felt that they are part of a “prestige club” after being accepted onto the running club exhibited higher motivation to continue with the program and improve their running performance. Additionally, when interviewing staff members and PA instructors, it seems that personal, almost individual instruction during PA classes (such as in the gym, where almost every participant had their own instructor, or during one-on-one swimming lessons) led to higher motivation among the participants and in turn, to higher adherence.

Relatedness that was found to be associated with well-being [30] may explain the improvement found in the current study, in the satisfaction and total score components from the Schalock Quality of Life Questionnaire, which assesses quality of life and well-being. It is possible that as the participants felt related to their peers in the activity group or to their trainers/coaches, as revealed in the interviews, they experienced improved satisfaction from life in general. This possibility is in line with previous results [31], whereby the ability to share feelings with “like-minded” others led to increased feelings of satisfaction among participants. These researchers also found that adults with ASD had a sense of well-being and belonging when they felt connections with others in their group. Our results are also in line with previous findings about many of the challenges faced by people with ASD in achieving well-being and belonging. It was found that the most frequent ways for people with ASD to achieve quality of life are by meeting personal needs within social settings, by achieving the impact of societal othering, by finding connection and recognition, and by managing relationships with friends and family [31].

The perseverance of the participants in the PA classes was also found to be associated with significant improvement over a six-month period in one fitness component, i.e., the two-minute walk test, from T0 to T1. These findings are in line with previous studies, whereby perseverance is associated with improved aerobic fitness following prolonged adherence to aerobic PA programs, such as swimming, running, or walking, among youth with ASD, where associations were seen between improved PA levels, skill-related fitness, muscular strength, and endurance [32]. In the current study, however, several fitness components did not improve during the 12-month intervention program, including muscular strength (measured via the standing long-jump), flexibility of the lower back and lower extremities (measured via the sit-and-reach test), and the mobility functional performance (measured via the TUG test). As such, future intervention programs could benefit from addressing these specific components in the PA classes that are offered to the participants.

Based on the qualitative analysis of the current study, it is assumed that there are several strength factors that enabled the impressive adherence of the participants. One of them is freedom of choice among PA modalities and schedules. The freedom of choice is of special importance; while external factors such as changes in physical appearance motivate people to begin exercising, enhancing their intrinsic motivation by freedom of choice is a key factor in promoting exercise adherence [33]. Moreover, freedom of choice might enhance participants’ sense of competence, autonomy, and relatedness attached to the physical activity they conducted, which in turn increases their adherence. Additional important factors that might have a positive impact on adherence are the dedication of the coordinator and the staff members, and a dedicated budget that enables the purchase of additional fitness equipment and the payment of staff salaries. Based upon the transtheoretical model [16], it is possible that this decision-making shaped the choices made in continuing, relapsing, or modifying PA behaviors by taking part in different PA modalities or schedules. However, it is essential to note that other factors that were not examined in the current study might also influence participants’ adherence to the program, such as the physical environment, social support, and support from caregivers [34]. Regarding this possibility, from the interview with the program coordinator, it emerged that many times, the participants’ decisions to join the physical activity training were influenced by the social workers. This fact seems to imply that the village managers and social workers should examine ways of encouraging independent decision-making by the participants.

In the interviews, the budget was mentioned both as a strength of the program (i.e., funding was achieving for the 12-month intervention program) and as a possible weakness and even threat of the program (i.e., new funding needs to be achieved in order to keep the program going). Similar factors were raised by Arnell et al. [35], who interviewed professionals from several aspects (education, health care, community, and sports organizations) about ways to promote physical activity in adolescents with ASD. Their results suggested that promoting PA habits in adolescents with ASD was difficult due to their need for tailored activities as well as other competing demands and limited resources. The program coordinator in the current study is determined to maintain this program, continuously seeking new sources of funding and additional volunteers—even on a weekly basic. In other words, a central component in the project’s success seems to be the dedication of the people who lead it. This finding highlights the importance of implementing a policy of funding health-promotional activities, such as the one proposed in the current study, by political bodies.

It is important to note that despite the general positive adherence that was found in the study, several issues still need to be addressed. For example, during the 12 months of the program, some participants did not attend any activity at all; thus, the coordinator and the social workers had to follow those participants more excessively. Additionally, there were two sub-groups of participants that exhibited lower levels of attendance: females (compared to males) and individuals with low functional levels (compared to those with medium or high functional levels). While these results are in line with previous studies [12], additional specific strategies are needed to enhance the attendance for those specific groups and decrease health risks among these populations.

The findings of this study are beneficial to all people involved in promoting programs for adults with ASD, including individuals themselves. However, several limitations should be addressed in interpreting and generalizing this study’s findings. First, the intervention program was conducted in a community village where the participants were recruited. It is possible that the results revealed in the current study are specific to this village. As such, future research could benefit from conducting similar intervention programs in several community villages, enabling greater generalization of the results. Next, the PA classes offered and their physical outcomes may not have been in line with the fitness tests conducted (such as the sit-and-reach test), and different tests may yield different or additional findings. The number of participants who took part in this study was relatively small, and as each participant was her/his own control, there was no “control group”. As such, future research could benefit from conducting a similar study on a larger scale of quantitative and qualitative participants. Another limitation pertains to the fact that participants had a free choice of selected physical activity types. As a result, it is difficult to evaluate the unique effects of each activity on physical development. Future studies, with bigger sample sizes, should aim at investigating the relative contribution of various types of activities. Finally, the fact that in the current study, there was no control group that did not exercise during the intervention period, may be an obstacle in generalizing the findings to the entire population of adults with ASD. However, it is important to note that as participants in the study were adults, natural improvements in the fitness parameters that showed improvement in the current study were not expected. Additionally, as the current population is known to have a high risk for deterioration in all fitness and health parameters assessed in the study, it is reasonable to assume that the improvements found in the current study were possible effects of the intervention. As it is not ethical to ask for a control group that will not exercise, future studies should compare the current intervention with other forms of physical activities (i.e., personal training versus group activities) and among people with ASD living in more open environments (and not in a specialized community village as was assessed in the current study). We also suggest exploring the current intervention among people with greater cognitive difficulties.

## 5. Conclusions

Based upon the findings of the current study, we may conclude that long-term PA interventions that are based on participants’ free choice to select between a range of PA classes, offered at convenient time slots during the day and promoted by significant people for the population, lead to increased PA adherence, improved aerobic fitness, and improved quality of life. Consequently, policymakers who plan health promotion programs for adults with ASD should aim at promoting long-term PA programs adjusted to this population and explore additional factors contributing to adherence to such programs.

## Figures and Tables

**Figure 1 healthcare-12-00247-f001:**
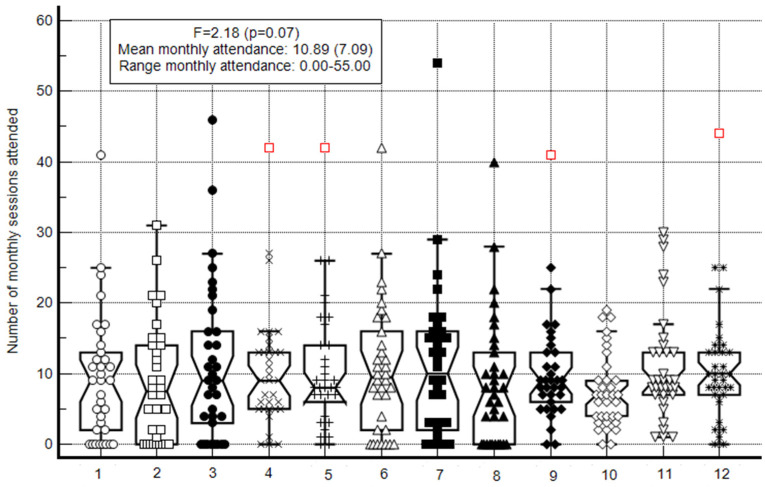
Monthly attendance Box-and-Whisker plot. Notes: The horizontal axis represents the activity month out of the 12-month program; The central box represents the values from the lower to upper quartile (25–75 percentiles); the vertical line extends from the minimum to the maximum value, excluding outside values which are displayed as separate points. An outside value is defined as a value that is smaller than the lower quartile minus 1.5 times the interquartile range, or larger than the upper quartile plus 1.5 times the interquartile range; the middle line represents the median.

**Figure 2 healthcare-12-00247-f002:**
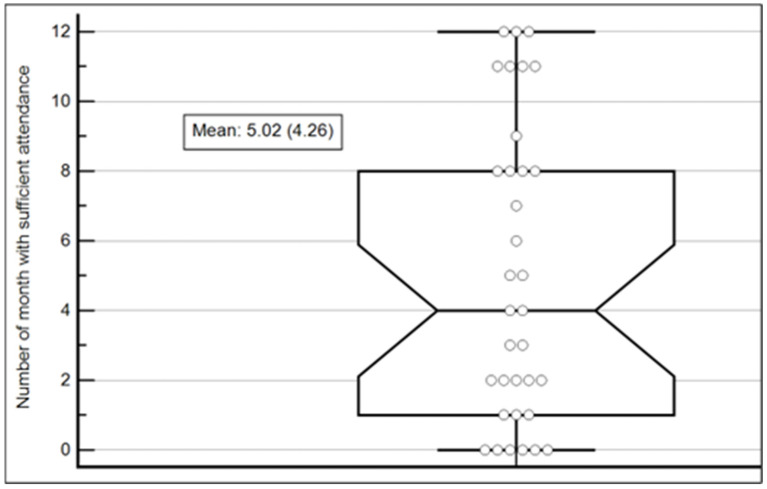
Number of months with sufficient attendance Box-and-Whisker plot. Notes: The horizontal axis represents the activity month out of the 12-month program.

**Figure 3 healthcare-12-00247-f003:**
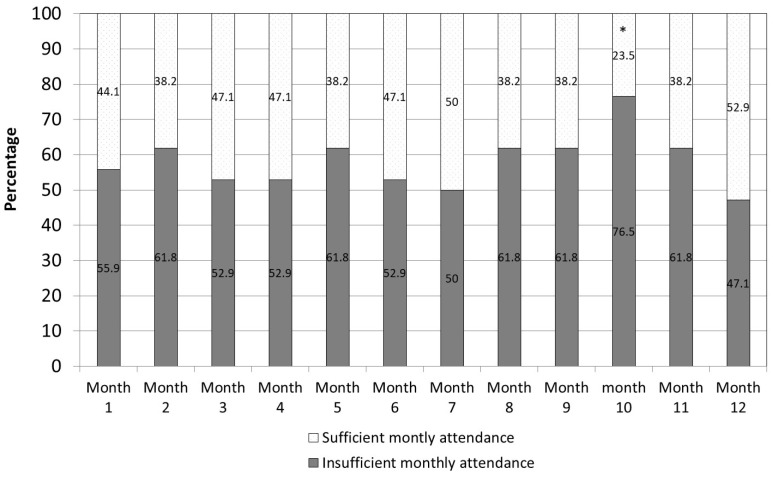
Percentage of participants according to sufficient/insufficient monthly attendance (n = 34). Notes: Using the Bonferroni correction for multiple comparisons, alpha < 0.004 (0.05/12) is considered significant.

**Table 1 healthcare-12-00247-t001:** Participants’ demographics, functional level, and health status (n = 34).

Variables			Mean (SD) [Range] OR n (%)	Chi-Squared (*p* Value)
Demographic characteristics	Age, years: mean (SD) [range]		39.76 (7.27)	------------
			[25.00–55.00]	------------
	Sex: n (%)	Males	24 (70.6)	5.76 (0.01)
		Females	10 (29.4)	
	Family status: n (%)	Single	30 (88.2)	46.29 (<0.001)
		Married	3 (8.8)	
		Divorced	1 (2.9)	
	Number of children: mean (SD) [range]		0.05 (0.34)	------------
			[0.00–2.00]	------------
	Educational level: n (%)	Elementary school	4 (11.80)	19.88 (<0.001)
		High school	30 (88.20)	
	Religiousness: n (%)	Secular	26 (76.5)	9.52 (0.002)
		Traditional	8 (23.5)	
ASD Functional level	Low: n (%)		8 (23.50)	2.17 (0.33)
	Moderate: n (%)		15 (44.1)	
	High: n (%)		11 (32.4)	
Health status	Stroke: n (%)	Yes	1 (2.94)	59.20 (<0.001)
		No	33 (97.05)	
	Epilepsy: n (%)	Yes	5 (14.70)	32.83 (<0.001)
		No	29 (85.29)	
	Respiratory diseases: n (%)	Yes	4 (11.76)	38.69 (<0.001)
		No	30 (88.23)	
	Cardiovascular diseases: n (%)	Yes	3 (8.82)	45.05 (<0.001)
		No	31 (91.17)	
	Hematological diseases: n (%)	Yes	1 (2.94)	59.20 (<0.001)
		No	33 (97.05)	
	Diabetes: n (%)	Yes	3 (8.82)	45.05 (<0.001)
		No	31 (91.17)	
	Body mass index: mean (SD) [range]		28.46 (5.32)	------------
			[21.31–41.00]	------------

Notes: ASD, autism spectrum disorderl SD = standard deviation.

**Table 2 healthcare-12-00247-t002:** Changes in fitness and quality-of-life measures.

Variables		T0: Mean (SD)	T1: Mean (SD)	T2: Mean (SD)	F-Ratio (*p*-Value)	Effect Size (T0 vs. T3)	Correlations (T0 and T2)
Fitness measures	Sit and reach: centimeters	15.06 (2.60)	14.78 (1.71)	15.35 (1.81)	0.06 (0.93)	0.12	0.66 ^†^
	Standing long jump: centimeters	84.63 (7.17)	78.18 (6.34)	80.19 (6.19)	0.07 (0.92)	−0.66	0.91 ^†^
	Two-minute walk test, meters	153.34 (6.91)	169.84 (7.03) *	164.84 (17.03) *	5.35 (0.001)	0.88	0.57 ^†^
Functional measures	Timed Up and Go, seconds	8.22 (2.97)	8.02 (2.77)	8.01 (2.60)	0.35 (0.70)	−0.07	0.80 ^†^
Quality-of-life measures	Satisfaction, score	24.48 (3.10)	27.09 (2.98) *	27.08 (2.38) *	5.30 (0.01)	0.94	0.22
	Productivity, score	25.59 (2.44)	26.21 (2.41)	27.14 (2.11)	0.07 (0.94)	0.67	0.17
	Power, score	23.96 (2.97)	24.09 (2.16)	24.33 (2.11)	0.10 (0.91)	0.14	0.36 ^†^
	Social, score	21.81 (4.08)	22.57 (3.45)	21.67 (2.09)	0.11 (0.90)	0.04	0.30 ^†^
	Total, score	95.71 (9.31)	99.84 (7.28) *	99.67 (6.28) *	5.25 (0.01)	0.49	0.47 ^†^

Notes: * Significantly different from T0 (*p* < 0.01, 2-tailed); ^†^ Statistically significant correlation (*p* < 0.05); light gray cells denote moderate effect size (effect size = 0.51–0.80); dark gray cells denote large effect sizes (effect size > 0.80).

**Table 3 healthcare-12-00247-t003:** Mean number of sufficient monthly attendance based on sex, religiousness, and functional level.

Variables		Mean Number of Sufficient Monthly Attendance: Mean (SD)
Sex	Males (n = 24)	5.87 (4.37)
	Females (n = 10)	3.0 (3.33) *
Religiousness	Secular (n = 26)	5.42 (4.09)
	Traditional (n = 8)	3.75 (4.84)
Functional level	Low (n = 8)	3.50 (3.96) ^b,c^
	Moderate (n = 15)	4.81 (4.30) ^a^
	High (n = 11)	6.00 (4.47) ^a^

Notes: * statistically significant differences between males and females (*p* < 0.05; 2-tailed); ^a^, statistically significantly different from “low” functional level (*p* < 0.05; 2-tailed); ^b^, statistically significantly different from “moderate” functional level (*p* < 0.05; 2-tailed); ^c^ statistically significantly different from “high” functional level (*p* < 0.05; 2-tailed); SD, standard deviation; sufficient monthly attendance was defined as attending at least 10 training sessions/month yielding at least 150 min of weekly physical activity; analysis was not conducted to “family status” and “educational level” on account of insufficient number of participants in “married” and “divorced” groups (n = 4) and “elementary school” group (n = 4).

## Data Availability

The data presented in this study are available on request from the corresponding author.

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
