# Peer review of "Health Promotion with Long-Term Physical Activity Program for Adults with Autism Spectrum Disorder"

_healthcare, 2024, doi:10.3390/healthcare12020247_

Round 1

Reviewer 1 Report

Comments and Suggestions for Authors

This is a well-written paper. The authors have identified research gaps with sufficient and relevant references. Both study purpose and hypothesis are based on reviewed literature and the transtheoretical model of behavior change. Additional strengths of this paper include the use of mixed methods approach and well-organized content.

Participants

·        Please explain how study participants were recruited and how the sample size was determined? Data analysis

Measurements

·        The description of each study variable is relatively brief.  What is the score range for each activity for physical fitness and functional ability measurements?

Data analysis

·        Was normality test of the key variables performed prior to performing parametric analyses (e.g., t-test, ANOVA)?

Results

·        Table 3. Were p values adjusted for multiple comparisons? 

Author Response

Thank you very much for the comments that helped improving the manuscript. Attached is our point-by-point response and corrections.

Reviewer 2 Report

Comments and Suggestions for Authors

Thank you for the opportunity to review this paper, reporting on an PA intervention to improve outcomes for adults with ASD. Overall the paper was clearly written and reports on an important area of policy and practice. 

Minor suggestions below: 

Please provide a little bit more explanation on the procedure/intervention itself and the context I.e. where were the PA classes conducted and by whom? Was this a disability specific context/village?  How was access to the classes funded? The latter is made clearer in the results, but would be useful in the methods. 

How do the QoL results of the cohort compare more broadly to ASD population and general population?

Unclear as to the relevance of religiousness as a measure? 

Results Line 235-237: please check Figure 2 explainer as it doesnt seem to align with figure? I.e. isnt there only one box in this figure and not one for each month? 

I wonder if Table 3 would be easier for readers if it was a simple bar graph?

Discussion: line 427, consider re-framing to encourage more opportunities for individuals with ASD to be involved in design and delivery of programs as this is an important rights-based approach and one that would quite likely lead to more feasible and sustainable options. Perhaps: the findings of this study are beneficial to all people involved in promoting programs for adults with ASD, including individuals themselves. 

Opportunities for a more financially sustainable intervention? Can you more strongly push for policy solutions to funding?

I would also want to see such an intervention made available to people living in more open contexts, i.e. no disability specific settings which I am presuming these community villages are? 

Need to unpack decision making theme. What can be done to support greater decision making of participants, both in relation to PA and more broadly. It is concerning that some participants have so little control over such important decisions. 

Future research considerations: given the overlap between ASD and cognitive difficulties, how could this intervention be adapted to also meet the needs of people with greater levels of cognitive difficulties. 

Considerations for next time: include a non-binary gender option for inclusion criteria. 

Author Response

(The authors gave the same response as above.)
